# The Physical Environment of Nursing Homes for People with Dementia: Traditional Nursing Homes, Small-Scale Living Facilities, and Green Care Farms

**DOI:** 10.3390/healthcare6040137

**Published:** 2018-11-26

**Authors:** Bram de Boer, Hanneke C. Beerens, Melanie A. Katterbach, Martina Viduka, Bernadette M. Willemse, Hilde Verbeek

**Affiliations:** 1Department of Health Services Research, Care and Public Health Research Institute, Maastricht University, 6200 MD Maastricht, The Netherlands; m.katterbach@alumni.maastrichtuniversity.nl (M.A.K.); m.viduka@alumni.maastrichtuniversity.nl (M.V.); h.verbeek@maastrichtuniversity.nl (H.V.); 2Opera Consultancy and Implementation, 5026 RK Tilburg, The Netherlands; hanneke.beerens@operaconsultancy.nl; 3Netherlands Institute of Mental Health and Addiction, Program on Aging, 3521 VS Utrecht, The Netherlands; BWillemse@trimbos.nl

**Keywords:** physical environment/space, nursing homes, small-scale living, green care farms, engagement, social interaction

## Abstract

It is well recognized that the physical environment is important for the well-being of people with dementia. This influences developments within the nursing home care sector where there is an increasing interest in supporting person-centered care by using the physical environment. Innovations in nursing home design often focus on small-scale and homelike care environments. This study investigated: (1) the physical environment of different types of nursing homes, comparing traditional nursing homes with small-scale living facilities and green care farms; and (2) how the physical environment was being used in practice in terms of the location, engagement and social interaction of residents. Two observational studies were carried out. Results indicate that the physical environment of small-scale living facilities for people with dementia has the potential to be beneficial for resident’s daily life. However, having a potentially beneficial physical environment did not automatically lead to an optimal use of this environment, as some areas of a nursing home (e.g., outdoor areas) were not utilized. This study emphasizes the importance of nursing staff that provides residents with meaningful activities and stimulates residents to be active and use the physical environment to its full extent.

## 1. Introduction

The importance of the physical environment for the well-being of people with dementia is well recognized. The ecological theory of aging and the environmental press model developed over 30 years ago stated that the fit between the environment and an individual’s cognitive and physical capacities is associated with the ability of people with dementia to age in place [1,2]. The built environment can avoid agitated or diffusing behavior, which might cause unnecessary harm [3,4]. Furthermore, it can support people with dementia to attain their full potential by positively influencing their autonomy, support their quality of life and well-being and attain the best possible potential of independence [4,5,6].

Literature reviews showed the importance of various environmental aspects for people with dementia (e.g., sunlight, sounds, view, spatial layout, nature, orientation, music, privacy, autonomy, windows, comfort, facilities, staff, group size, non-institutional character, and domesticity) [6,7,8]. Especially for people with dementia, the environment supports the physical and cognitive requirements of an individual, implying the importance of a balance between the person and the environment. Studies suggest that it is recommended to build nursing homes of smaller size with an open-plan environment in which visual access is favored. These support orientation and social interaction, and facilitate caregiving for nurses, as residents can be located more easily [9]. Sensory stimulation should also be taken into consideration. On the one hand, it is important that stimulus reduction features are considered to assure that residents are not overwhelmed by too much information (environment press) or noise. On the other hand, the design should highlight useful stimuli such as familiar cues to bathrooms and exits to safe outside areas [9]. Another essential design feature of a nursing home is to create an atmosphere of familiarity with elements of the past, as this is what people with dementia most easily recall. It is also important to consider privacy by providing private spaces, in which residents can be alone or in close company of a friend. Moreover, public spaces for community activities and other social interaction are important. It is also of great interest to support people with their activities of daily living, to allow them to have their own routines and to provide a homelike atmosphere [10].

The substantial evidence of the role of the physical environment for people with dementia affects the nursing home care sector. There is increasing interest in the design of the physical nursing home environment and how this supports person-centered care [5,11,12,13]. For instance, a homelike environment positively influences residents’ daily activities and social interactions [14]. Advances in the nursing home care sector focus on the development of small-scale, homelike care environments such as green care farms (GCFs). GCFs provide care for people with dementia in a small-scale homelike facility in which a familiar atmosphere and normal daily living is emphasized. People with dementia have the opportunity to engage in activities with, e.g., crops, livestock and woodland, in which they can make use a unique physical environment consisting of several areas on the farm such as the kitchen, shed, gardens, farmyard, and stables. Freedom of movement is emphasized and giving people with dementia autonomy on their own lives (and the choices they make) is a central part of providing care at GCFs [15,16,17].

Consequently, a shift from a traditional medical model towards a psychosocial homelike model of care takes place [18]. Instead of long corridors and shared rooms, nursing homes are increasingly small-scaled and homelike with a familiar physical environment. Hence, a sense of at-hominess is created by providing meaningful experiences of choice, mastery, and social interactions [11,18].

Several instruments have been developed to map the physical environment of a care facility [9,19,20,21,22]. These instruments include aspects such as maintenance, cleanliness, safety, lighting, domesticity, noise, and familiarity. However, these measurement instruments are rarely focused on people with dementia and often focus on traditional medical environmental aspects, such as the presence of safety bars and slippery floors. Hence, they do not comprehensively assess all factors of importance to well-being of people with dementia. Most studies that compare different types of nursing home environments provide a general description of the physical environment (e.g., large-scale versus small-scale), and focus on measuring generic, broad outcomes such as quality of life, and quality of care outcomes such as falling incidents and medication use [23,24]. Furthermore, only few studies investigate whether differences in environmental aspects between nursing homes lead to benefits for nursing home residents with dementia in terms of their daily life (e.g., activities or social engagement). Some studies suggest that a high-quality care environment leads to residents that are more active, engaged, and have a better quality of life [10,12,13,25]. However, research on how a physical environment is used by residents is scarce.

In this paper, two studies are described. First, an evaluation of different care environments was carried out using the OAZIS-Dementia, an assessment tool specifically developed for the Dutch nursing home context. It was investigated whether there are differences in terms of the physical environment between traditional nursing homes, small-scale living facilities, and green care farms. Second, a study was conducted in which how different nursing home environments are used by their residents was assessed. 

## 2. Materials and Methods 

### 2.1. Study Design

Both studies used a cross-sectional observational study design, and were part of a larger research project investigating the effects of green care farms for people with dementia [15]. This study was declared not to be invasive for people with dementia by the medical ethics committee of the Maastricht University Medical Centre (14-05-003).

### 2.2. Study 1

#### 2.2.1. Setting

Two types of nursing homes for people with dementia were compared: traditional nursing home wards and small-scale living facilities. The latter consisted of three subtypes: (1) stand-alone small-scale living facilities; (2) small-scale living facilities on the terrain of a larger nursing home; and (3) green care farms. Table 1 provides a brief description of each type of nursing home. In total, the physical environment of 18 nursing home wards was mapped.

#### 2.2.2. Instruments

The OAZIS-Dementia was developed to measure the physical environment of long-term care environments in a Dutch setting [15]. During the development of the OAZIS-Dementia, face validity and content validity were taken into account in various ways. Existing literature and instruments [6,7,8,9,20,26,27] were reviewed systematically by two researchers to investigate whether the items were relevant for Dutch nursing homes for people with dementia. 

Subsequently, the relevance of the theme items was discussed with experts in nursing home care (care professionals, researchers, real-estate controllers, and location managers). During these discussions, the categories of the OAZIS-Dementia and the specific items were addressed in detail. A pilot test in three nursing homes during the development of the instrument showed that the inter-rater reliability of the OAZIS-Dementia was high, with an ICC of 88.

The OAZIS-Dementia consists of 72 items, which assess aspects of the environment on a five-point Likert scale, ranging from 1 (not at all) to 5 (completely). The checklist is divided into seven categories that emerged from reviewing the existing literature and existing instruments: (1) privacy and autonomy; (2) sensory stimulation; (3) view and nature; (4) facilities; (5) orientation and routing; (6) domesticity; and (7) safety. Higher scores indicate a higher probability for the environment to have a positive effect on its residents. Table 2 summarizes the categories measured with example items that were scored for each nursing home. Each item in the OAZIS-Dementia has the same weight in terms of calculating total scores. The OAZIS-Dementia is available upon request.

#### 2.2.3. Procedure

Two researchers (who were involved in the development of the OAZIS-Dementia) visited all wards for another observation study several times. The researcher, who visited a ward most frequently, filled out the OAZIS-Dementia for that specific ward. The OAZIS-Dementia was filled out during the third or fourth visit, so that the researcher was already familiar with the environment. It took approximately 1 h to fill out all items. The designated scores were reviewed by the other researcher and, in the case of disagreements, discussed. 

#### 2.2.4. Data Analysis

All 72 items were scored on a five-point Likert scale. For each category, an average value was calculated by adding the item scores and dividing them through the number of items. A final score on the OAZIS-Dementia was calculated in the same manner. Descriptive statistics were used to check for differences between the types of nursing homes. 

### 2.3. Study 2

#### 2.3.1. Setting

Three nursing homes were included in this study, which all have been purposefully built according to the principles of small-scale, homelike care environments. All three nursing homes can be categorized as a small-scale living facility on the area of a larger nursing home (from the categorization of Study 1). Below, a more specific description per nursing home is given.

Nursing Home 1 (NH1) had six single standing residential units with eight residents with dementia in each. The units were accessible individually via an entrance door either on street level or a stairway. The three buildings were purposely built as archetypal houses. Every resident had his or her own bedroom (215 ft^2^) including a bathroom, shared with a resident on the opposite side. Space for staff was organized in the entrance area for privacy and confidentiality of the residents. Nevertheless, staff took their breaks within the common spaces of the residents.

Nursing Home 2 (NH2) had six residents per unit designed specifically for people with dementia. Every resident had his/her own bedroom, with a room (190 ft^2^) including a sink. Two bathrooms were shared amongst the six residents. Nursing staff had no private or separate space. Spaces for nursing equipment or exits to leave the unit were not freely accessible for residents. The main ground floor of the facility accommodated a hairdresser, restaurant, physiotherapist, reception area, and offices for managerial and administrative work and an accessible enclosed outdoor garden including an animal shelter. 

Nursing Home 3 (NH3) incorporated 71 apartments, of which 32 were occupied by residents with dementia and 39 by residents with somatic disorders. Each resident had their own unit (450 ft^2^) consisting of a kitchen, bedroom, and private bathroom, furnished with familiar belongings from the residents’ previous homes. In addition to private apartments, the units had a communal kitchen/dining area and large living area to share with another unit on the same floor occupying another eight residents. On ground level, there were administrative offices for management or nursing and medical personnel, and a physiotherapist practice. The facility was built in the countryside surrounded by other apartment complexes. Outside, garden areas were accessible by residents accompanied by family, friends or personnel.

#### 2.3.2. Instruments

To identify environmental features of the different settings in the study, the OAZIS-Dementia was used in each setting. Additionally, there were two 10-h observations per nursing home (8.30–18.30), composed of one-day shift and one-evening shift. Night shifts were deliberately excluded, as residents were assumed to sleep during this timeframe. During these observations, the extent to which residents used the physical environment was observed. A subset of the aspects of daily life observed with the Maastricht Electronic Daily Life Observation tool (MEDLO-tool) was used [28]. The MEDLO-tool is a tablet-based observational tool that assesses aspects of daily life. 

The following aspects of daily life were observed: (1) the engagement in an activity (yes/no); (2) the location where an activity occurred (4 options); and (3) the social interaction (type of social interaction, and with whom). Table 3 gives an overview of the aspects that were observed to map the usage of the physical environment in terms of daily life. The MEDLO-tool was demonstrated to be a valid, feasible and reliable observation tool with high absolute agreement (86%) between observers and Kappa values between 0.5 and 1.0. Thus, the MEDLO-tool has good psychometric properties [28].

#### 2.3.3. Procedure

The researchers who were involved in data collection for Study 2 received a short training on how to use the OAZIS-dementia and the MEDLO-tool. The training consisted of studying the instruments and their manuals, and discussing these with the main researchers (who were involved in developing both tools). Example situations were discussed to make sure observers would score the same situation in the same manner. These discussions were also carried out during data collection.

Furthermore, for this study, the observation procedure of the MEDLO-Tool, and the aspects observed were slightly altered, due to practical reasons (available time/resources), and the aim of the study (most relevant aspects of daily life were chosen). Residents were observed for 1 min each on a randomized basis. Each resident was observed during a 1-min “snapshot” before moving on to the next resident, until all residents with dementia residing in the small-scale unit at the time of observation were captured. After 20 min, the first observation round was finished, filling in all items of the MEDLO-tool. This procedure was repeated on six observation days for a 10-h observation shift (08:30–18:30). Every 2 h, observers took a 30-min break. 

#### 2.3.4. Data Analysis

First, the OAZIS-Dementia scores were calculated as in Study 1. Second, descriptive analysis on the aspects of daily life was conducted. For engagement and social interaction, percentages were calculated. A percentage thus indicated the proportion of the observations that a resident was engaged in an activity, or had social interaction. For the other aspects that were observed (location, type of social interaction, and social interaction with whom), the percentages of the individual scoring options were calculated.

## 3. Results

### 3.1. Study 1

#### Comparison between Types of Nursing Homes

Table 4 shows the mean scores on each category of the OAZIS-Dementia per nursing home type. Furthermore, a total score is given. Lowest values are presented in orange and highest values in green. In general, all types of small-scale, homelike care environments score better on environmental aspects compared with traditional nursing homes, especially green care farms. Green care farms have high scores on most categories (privacy and autonomy, view and nature, orientation and routing, and domesticity), resulting in the highest total score as well. 

Traditional nursing homes have the lowest values on almost all categories (privacy and autonomy, sensory stimulation, view and nature, orientation and routing, and domesticity), resulting in the lowest final total score. The stand-alone small-scale living facilities have the lowest on the facilities category. Small-scale living facilities on the terrain of a larger nursing home have the highest score on sensory stimulation and facilities. No differences were found for the safety category across the nursing home types.

### 3.2. Study 2

#### 3.2.1. Comparison between nursing homes

Table 5 presents the outcomes of the OAZIS-Dementia assessment for each nursing home. All nursing homes scored above 3 on every item, indicating high overall scores for each nursing home. Minimal differences were found on the total scores (4.1, 3.9, and 4.1). Largest differences were found on the categories of privacy and autonomy, and domesticity. Especially, the domesticity items include not only physical environmental aspects (e.g., homelike appearance) but also items on organizational environmental aspects, such as whether residents can decide the time they want to get up and go to bed. 

#### 3.2.2. Use of the Physical Environment

In total, 2043 observations were conducted, 807 observations in NH1, 524 in NH2, and 712 in NH3. The number of six residents living in this facility can explain the comparatively lower number of observations in NH2. The other nursing homes accommodate eight residents per unit, resulting in a higher number of observations. 

Table 6 provides an overview of where residents spent their time during the observations, how often they were engaged in an activity, and whether they had social interaction. Residents of NH1 spent 54% in communal areas. Residents directly found themselves in different communal areas upon leaving their bedrooms. In contrast, resident rooms of NH2 and NH3 were located along the hallways. Overall, residents of NH2 spent most time in communal areas (78%, see Table 6), and residents of NH3 the least (40%). Private rooms furnished with own belongings, which were recognizable for residents, were used more often. This was observed in NH1 (34%) and in NH3 (57%) where residents had their own apartments with different housing areas (kitchenette, living room, bedroom, bathroom). Residents of NH2, which had the least homelike bedroom and the least volume in space, spent 9% of their time in private space, over the course of observations. NH1 had an outdoor patio, which was used in 8% of the observations. The balcony of NH2 was used in 4%. Easily accessible balconies of NH3 have not been observed to be utilized by residents (see Table 6).

When activities took place, residents mostly engaged in that main activity. Participation was observed to be highest in NH1 (92%), followed by NH3 (87%). NH2 had the least engagement in activities with 82%. When residents were not engaging in main activities, they were engaged with something else, gazing, or sleeping. Residents often fell asleep at the dining tables after mealtime.

Most social interaction was observed for NH1 (54%), followed by NH2 (52%), and NH3 (37%). Residents in all three nursing homes spent most of their time interacting with staff within the communal areas that were observed. Those in NH1 had more interaction with other residents than the other two nursing homes. All nursing homes had mostly positive social interactions. In all nursing homes, the amount of interaction with family, friends or others was very low (<5%).

## 4. Discussion

Results of the current study indicate that the physical environment of small-scale living facilities for people with dementia has more potential to be beneficial for residents’ daily life than the physical environment of traditional large-scale nursing homes. Traditional nursing homes did not facilitate privacy and autonomy, sensory stimulation, view and nature, orientation and routing, and domesticity. However, this study also found that having a potentially beneficial physical environment does not automatically lead to an optimal use of this environment. Specific areas of a nursing home (e.g., the outdoor area) were not utilized. Nursing staff appeared as an important factor for whether the potential of the space was used.

Linking the physical environment to outcomes concerning daily life is important to investigate the person–environment fit (P-E fit). Small-scale, homelike nursing homes may have a better P-E fit for residents living with dementia [29] as they promote activity engagement and quality of life [30]. Matches are needed among a person’s needs, his/her abilities, and environmental demands to support positive outcomes such as a higher well-being, better nutrition, less medication, and more person-centered care [31,32,33]. However, the P-E fit may decrease for residents when the dementia progresses and environmental demands may exceed functional abilities, resulting in lower activity engagement [34]. This study found that especially green care farms adopt a positive physical environment for residents with dementia. In another study, we found that residents of green care farms displayed a more active daily life, were more socially active, came outside more often, and were more actively engaged than residents in traditional nursing homes [25]. These results suggest that the positive environmental components of green care farms may positively impact their daily life [25,35].

Results of this study suggest that nursing staff can be of importance for stimulating the optimal use of a stimulating physical environment. In alignment with the ecological theory of aging, activity involvement, high quality of life, and well-being for residents can be achieved by adjusting/tailoring activities to different coping capabilities of older adults. Therefore, nursing staff should consider individual preferences, and cognitive and physical conditions [30]. Moreover, interaction and engagement by staff with residents foster a person-centered care approach [36,37,38], can arouse cognitive abilities of people with dementia [39], and provide a meaningful use of the physical environment. Therefore, staff are decisive for the use of different areas more purposively [40]. 

There is also a need for nursing staff to adapt their work to encourage residents to participate in daily activities in their nursing home [38]. If the built environment can support this adaptation, the likelihood of nurse encouragement may increase. For the staff working in an environment with smaller facilities, tasks are more integrated and less specialized than in traditional wards [15]. In these small-scale environments, nursing staff have responsibility not only for essential nursing tasks such as medication administration and personal care, but for food preparation, housekeeping, and social and recreational activities as well [12]. Providing an environment supportive to the nursing staff, which accounts for time constraints and workload in small-scale living nursing homes is critical. 

The built environment can play a significant role in supporting nursing staff in integrating resident engagement into their daily nursing tasks. A recent study by Lee, Chaudhury and Hung (2016) explored staff perceptions on the role of the physical environment in dementia settings. Staff felt that being close to residents such as in a small-scale living space provided familiar positive stimulation that empowered them to connect with the residents [41]. 

Continuing to participate in activity is vital to the quality of life of nursing home residents and nursing care should include assisting residents with this participation. In the study, this was accomplished in the nursing homes that had a supportive built environment through open, large rooms, with visual access to each other and appropriate, comfortable seating. Additionally, in the open kitchen/dining rooms, positive sensory stimulation was created; for example, when nursing staff were preparing food, the smells and sounds of cooking could be sensed throughout the home, which may encourage residents to gather. These features of small-scale living made it easier for the nurses and residents to be together in the communal areas. This is in line with a recent review showing that the physical environment can be linked with therapeutic goals for people with dementia [21]. The authors of this review indicated that certain facility characteristics such as unit size, spatial layout, or having an outdoor area can be linked with therapeutic goals such as maximizing awareness and orientation, support functional abilities, and social contact [21].

### Limitations and Recommendations for Future Research

Some methodological considerations should be taken into account. First, the study had an explorative, descriptive character, including a small number of participating nursing homes, which limits the generalizability of results. Second, this study used mainly a quantitative approach for data collection on activity involvement and use of space. Collection of qualitative data for example by interviewing residents, family members or nursing staff would gain valuable information on why certain spaces were used less or more and how the environment was experienced. One limitation is that information regarding cognitive status and functioning levels across the three nursing homes is missing. Although the nursing homes have similar admission criteria, it is difficult to determine how comparable the residents across these nursing homes were. This could have affected the differences that were found in terms of the use of the physical environment. Future studies should make sure that observational data can be compared with information regarding cognition and functional status of individual residents. The OAZIS-dementia instrument used in this study has some limitations that are in line with other observational instruments to measure the physical environment. It is a relatively long instrument to fill out. Furthermore, it is beneficial if a researcher has visited the nursing a couple of times before answering all the items (which makes it more time consuming). Lastly, it remains difficult to ascertain which aspects of the environment are associated with specific outcomes for residents due to the interrelationships of the organizational, social, and physical environment [21]. Future studies should focus more on specific relationships (e.g., by manipulating a certain part of the environment).

## 5. Conclusions

The physical environment of small-scale, homelike nursing homes has more potential to be beneficial for people with dementia than traditional nursing homes. However, the environment is still not utilized to its full potential, which can affect the engagement in activities and social interactions of people with dementia living in a nursing home.

## Figures and Tables

**Table 1 healthcare-06-00137-t001:** Description of the types of nursing homes.

Type of Nursing Home	Brief Description	Prominent Characteristics of the Physical Environment
Traditional nursing home ward	≥20 residents on the ward Differentiated tasks for staff Routines and rules of the organization determine daily life	Large building, long corridors, shared rooms, hospital-like atmosphere, separate kitchen, facilities such as a restaurant and activity areas are attached to the ward
Small-scale living facility on the area of a larger nursing home	Maximum of 8 residents Joint household Meals (including dinner) prepared inside the home three times a day Integrated tasks for staff Small team of caregivers Residents and informal caregivers determine daily life	Homelike situation, single rooms, familiar interior, common living room attached to kitchen, facilities such as a restaurant and activity areas are attached to the ward, outdoor area accessible
Stand-alone small-scale living facility	Has the same characteristics as a small scale living facility on the terrain of a larger nursing home, however situated in a neighborhood Aims at close connections with the community and opportunities to maintain a social network.	Archetype house, single rooms, familiar interior, common living room attached to kitchen, no direct access to facilities provided at a larger nursing home, outdoor area accessible
Green care farms	A type of stand-alone small-scale nursing home facility in a rural area Both care and agricultural activities are important. House on the area of the farm.	Homelike situation, archetype house, single rooms, familiar interior, common living room attached to kitchen, freely accessible outdoor areas, stables, gardens, animals

**Table 2 healthcare-06-00137-t002:** OAZIS-Dementia categories and example items.

Category	Item No.	Examples
Privacy and Autonomy	Item 1–7	Residents have a single room Washrooms are discrete
Sensory Stimulation	Item 8–25	Daylight glare and harsh reflections are prevented or can be individually regulated with blinds Staff can regulate temperature
View and Nature	Item 26–36	Residents have views of nature and greenery There are animals present
Facilities	Item 37–45	The outdoor area is accessible for people using a wheelchair or walker There are several spatial facilities on the ward to meet other residents
Orientation and Routing	Item 46–52	The structure of the ward is open Use of clear icons/nameplates to denote toilet and bathroom
Domesticity/Small Scale	Item 53–69	The ward has its own front door with a doorbell The staff does not wear uniforms
Safety	Item 70–72	There are devices dedicated to security present at the toilets Floors are not slippery

**Table 3 healthcare-06-00137-t003:** Scoring options during observations.

Aspects of MEDLO-Tool	Operationalization	Scoring Options
Engagement in activity	Five category options	Yes, active engagement (participating in activity) Yes, passive engagement (focus on activity) Yes, engagement with something else No, not engaged (gazing without focus) No, not engaged (sleeping)
Location	Five category options	Communal area on the ward Own room Communal area off the ward Outside
Level of social interaction	Five category options	No social interaction No social interaction, resident attempts to interact, gets no response No social interaction, environment attempts to interact, but resident does not respond Yes, interaction with someone else Yes, interaction with two or more people
Type of social interaction of environment towards resident	Five category options	Negative restrictive (interaction that oppose or resist resident’s freedom of action without good reason, or ignore resident as a person) Negative protective (providing care, keeping safe or removing from danger in a restrictive manner without explanation or reassurance) Neutral (brief, indifferent interactions) Positive care (interactions during the appropriate delivery of care) Positive social (interactions principally involving “good, constructive, beneficial” conversation and companionship)
Social interaction with whom	Five category options	Staff Other residents Family and/or friends Others Combination of the above

**Table 4 healthcare-06-00137-t004:** Scores on the OAZIS-Dementia per type of nursing home.

OAZIS-Dementia Categories	Traditional Nursing Home Ward (n = 4)	Small-Scale Living Facility on the Terrain of a Larger Nursing Home (n = 6)	Stand-Alone Small-Scale Living Facility (n = 3)	Green Care Farm (n = 5)
Privacy and autonomy	2.8	4.0	4.7	4.7
Sensory stimulation	3.5	4.4	3.7	4.2
View and Nature	2.9	3.6	3.0	4.3
Facilities	3.6	4.2	3.3	3.7
Orientation and routing	2.5	3.6	3.7	3.8
Domesticity	2.1	4.2	4.3	4.5
Safety	4.3	4.4	4.3	4.3
Total	3.0	4.1	3.8	4.2

**Table 5 healthcare-06-00137-t005:** Scores on the OAZIS-Dementia per nursing home.

OAZIS-Dementia Categories	Nursing Home 1	Nursing Home 2	Nursing Home 3
Privacy and autonomy	4.9	4.3	5.0
Sensory stimulation	3.8	3.8	4.2
View and nature	3.6	3.6	3.8
Facilities	4.4	4.0	4.3
Orientation and Routing	3.6	3.9	3.3
Domesticity	4.2	3.3	3.7
Safety	4.3	4.3	4.7
*Total*	4.1	3.9	4.1

**Table 6 healthcare-06-00137-t006:** Percentages on location, engagement and social interaction.

Category		Nursing Home 1	Nursing Home 2	Nursing Home 3
Location	Communal area on the ward	54%	78%	40%
	Own room	34%	9%	57%
	Communal area off the ward	4%	9%	3%
	Outside	8%	4%	-
Engagement in an activity		92%	82%	87%
Social interaction		54%	52%	37%
Social interaction with whom	Staff	35%	37%	49%
	Other residents	29%	15%	13%
	Family and/or friends	1%	5%	1%
	Others	9%	12%	11%
	Combination of the above	26%	32%	27%
Type of social interaction	Negative restrictive	1%	-	-
	Negative protective	1%	1%	2%
	Neutral	8%	16%	24%
	Positive care	39%	25%	24%
	Positive social	52%	59%	50%

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
