# Peer review of "The Physical Environment of Nursing Homes for People with Dementia: Traditional Nursing Homes, Small-Scale Living Facilities, and Green Care Farms"

_healthcare, 2018, doi:10.3390/healthcare6040137_

Reviewer 1 Report

This study adds to the growing body of evidence-based literature on the role of the physical environment in dementia care settings. The study's strength lies in the comparative analysis of different types of small scale facilities, care farms and traditional care facilities. The paper needs be improved on several areas to make it a worthwhile contribution. 

1.     The literature review needs to highlight findings from the existing research on green care farms. Although references on green care farms are included, their merits as a distinctive small scale facility needs better articulation. 

2.      A comprehensive review on the role of the built environment in dementia care facilities was recently published by Chaudhury et al in the Gerontologist journal, which needs to be included in the review section of this paper and additional issues incorporated.

3.     The introduction section needs to better articulate the rationale for this comparative study. There have been several publications based on studies comparing traditional care homes with small scale homes (e.g., the US based studies comparing the Greenhouse model with traditional facilities). That body of work needs to be included along with a clear justification for this comparative study,

4.     What was the basis of the 7 categories of OAZIS-Dementia? Existing environmental assessment tools? Empirical literature? Is there any variation of salience across the 7 categories or they equally important in this tool?

5.     The three nursing homes in study two are not clearly distinguished. It seems that the units in NH1 and NH2 are quite similar. A table would be helpful to clearly present the differences across the three types. Nevertheless, this reviewer is not convinced there is enough difference between NH1 and NH2. These two types could be merged as one type. 

6.     No psychometric properties are included for the MEDLO-tool. What is the reliability or validity information. There is some potential overlap between two aspects of the MEDLO-tool: Engagement in activity and Level of social interaction. In the event of a planned activity that entailed social interaction with an activity leader and/or other residents, those instances could count under both of those two aspects. 

7.     One critical information piece missing in this paper is the cognitive status and functioning levels across the three care homes. How comparable (or not) are the residents across the three sites? This would have direct implication on the use of the communal spaces. E.g., were the residents in NH3 higher functioning compared to NH1 and NH2?

8.     The discussion section is based on past literature and is somewhat superficial in taking a critical look in interpreting the findings of the study. E.g., Why did the residents in NH2 spent the most time in the communal areas compared to the residents in NH1 and NH3? Is it because they were not able to get back to their own rooms independently? 

9.     The discussion section can be strengthened with a clearer articulation of the interrelationship between the physical and social environments. Although the issue is mentioned, this needs to be better supported by the data.

10.  Recommendations for future research should indicate the need for comparative analysis on resident outcomes beyond use of space and social engagement, e.g., anxiety, agitation, walking across the different types of small scale nursing homes.

Author Response

A response to the comments can be found in the uploaded word file.

Reviewer 2 Report

This study is an important contribution to the knowledge about the influence of different aspects of the physical environment of dementia care settings on people with dementia. The research is well-executed and clearly written. Some revisions could be made to improve the paper.

Line 69  The author could also mention the reviews of Chaudbury et al,(2017) and Elf et al,(2017) in this paragraph.

Line 103  On which existing literature and instruments is the OAZIS-dementia based on?

119.  Some clarification regarding the use of the instrument is needed. How much time does it take to fill out the OAZIS –dementia on average? Does the researcher received training before conducting the instrument? After how many visits was the instrument filled out by the researcher. Was this the same number of visits for every ward?

Line 133 and paragraph 3.2.1  To provide more insight in the relation between study 1 and study 2 the author could specify the type of nursing homes according to the types listed in table 1.

Line 155  For these observations a different observation procedure was followed then mentioned in the article about the development of the MEDLO tool: de Boer et al, (2016). Why was choices for the different procedure?

Limitations  Could the author also mention some limitations of the OAZIS-dementia instrument? In the earlier mentioned literature reviews some limitations of current instruments are mentioned. Does the OAZIS-dementia resolve some of these limitations?

Author Response

(The authors gave the same response as above.)

Reviewer 3 Report

The authors compared the physical environment between three types of nursing homes.

As they mentioned in limitations, the study is explorative and descriptive, yet is interesting with focus on psychosocial environment as well as medical aspects.

Some revisions would be extended to:

1. Definition of ‘green care farms’ in 1. Introduction

2. Whether researchers who conducted evaluation using specific tool have received training for the evaluation in 2.1.3. Procedure and 2.3.3. Procedure

3. How the Study 2 corresponded to the Study 1 in 2.3.1. Setting (three nursing homes in Study 2 were also included in Study 1 or not)

4. From what parts of results indicate that nursing home is essential in 4. Discussion, third paragraph

Author Response

(The authors gave the same response as above.)
